# Rethinking nnU-Net for Cross-Modality Unsupervised Domain Adaptation in Abdominal Organ Segmentation

Ziyang Chen[1,2], Xiaoyu Bai[1,2], Zhisong Wang[1,2], Yiwen Ye[1,2], Yongsheng Pan[1,2,3(✉)], and Yong Xia[1,2,3(✉)]

[1] Research & Development Institute of Northwestern Polytechnical University in Shenzhen, Shenzhen 518057, China
[2] National Engineering Laboratory for Integrated Aero-Space-Ground-Ocean Big Data Application Technology, School of Computer Science and Engineering, Northwestern Polytechnical University, Xi'an 710072, China
[3] Ningbo Institute of Northwestern Polytechnical University, Ningbo 315048, China
{zychen, bai.aa1234241, zswang, ywye}@mail.nwpu.edu.cn,
{yspan, yxia}@nwpu.edu.cn

**Abstract.** Research on abdominal organ segmentation has been extensive for computed tomography (CT) scans but limited for magnetic resonance (MR) scans due to the scarcity of annotated MR data. This challenge highlights the need for effective cross-modality unsupervised domain adaptation (UDA) techniques to leverage annotated CT scans for improving MR scan segmentation. While nnU-Net is recognized as a robust baseline for medical image segmentation, its application in UDA has been underexplored. In this paper, we propose a novel approach that rethinks nnU-Net as a tool to enhance UDA methods for abdominal organ segmentation in MR scans. We introduce a three-stage pipeline to address this challenge. In the first stage, we develop an nnU-Net-based UDA framework with a triple-level alignment strategy to facilitate knowledge transfer from CT scans to MR scans. In the second stage, we use the nnU-Net trained in the first stage to generate pseudo labels for MR scans. We then fine-tune this model with both labeled CT scans and MR scans with pseudo labels, and additionally train a separate nnU-Net from scratch using the pseudo-labeled MR scans. In the third stage, we address resource constraints by training a lightweight nnU-Net with selected unlabeled MR scans and their corresponding pseudo labels. We evaluate our approach on Task 3 of the FLARE2024 challenge, where the lightweight nnU-Net achieves a mean Dice Similarity Coefficient (DSC) of 75.37 and a mean Normalized Surface Dice (NSD) of 81.67 on the validation set. Our code is publicly available at https://github.com/Chen-Ziyang/FLARE2024-Task3.

**Keywords:** Cross-modality · Unsupervised domain adaptation · Abdominal organ segmentation.

## 1   Introduction

Abdominal organ segmentation, which involves delineating anatomical structures from medical images, is essential for various clinical applications. While segmentation in computed tomography (CT) scans has seen significant advancements over the past decade [6,14], progress in magnetic resonance (MR) scans remains limited due to the scarcity of annotated MR data. In contrast, a large volume of annotated CT scans is available, and acquiring numerous unlabeled MR scans is relatively straightforward. This discrepancy underscores the need for cross-modality unsupervised domain adaptation (UDA) to utilize annotated CT scans and enhance MR scan segmentation performance [19].

Recent advancements in deep learning have facilitated automatic segmentation of multiple abdominal organs [3,4]. Among these methods, nnU-Net [9] stands out as a prominent baseline, underpinning many leading solutions in medical image segmentation challenges [11,7,10]. nnU-Net's success is largely due to its self-configuring nature, which adapts model architecture, data preprocessing, and training strategies to the dataset's specific characteristics. This adaptability makes nnU-Net a popular choice with minimal manual intervention. However, nnU-Net's default configuration does not accommodate tasks involving unlabeled data, limiting its use in UDA scenarios.

In this paper, we propose a novel approach to adapt nnU-Net for cross-modality UDA to enhance abdominal organ segmentation in MR scans. We introduce a three-stage pipeline designed to optimize both effectiveness and efficiency. In the first stage, we develop a nnU-Net-based UDA framework inspired by [2], incorporating input-, feature-, and output-level alignment to transfer knowledge from CT to MR scans, resulting in a universal segmentation model (Uni-Net). We train a conditional generator based on CGAN [15] for input alignment to translate CT scans to MR-like images. For feature alignment, we disentangle and align style and content features using consistency constraints. At the output level, we apply a segmentation consistency constraint to ensure uniformity across modalities. In the second stage, we leverage the Uni-Net to generate pseudo labels for MR scans. We then fine-tune the pre-trained Uni-Net using both labeled CT scans and MR scans with pseudo labels. Additionally, we train a new nnU-Net from scratch, referred to as MR-Net, using MR scans with pseudo labels. In the third stage, we use the fine-tuned Uni-Net and MR-Net to generate and select robust pseudo labels based on consistency. To address resource constraints, we develop a lightweight nnU-Net, called LW-Net, which is trained on the selected unlabeled MR scans and their corresponding pseudo labels following nnU-Net's default configurations. We conducted experiments on Task 3 of the FLARE2024 challenge, which involves 2,050 labeled CT scans and 4,818 unlabeled MR scans. We evaluated all the models on the validation set comprising 110 MR scans, and our proposed LW-Net achieved a mean Dice Similarity Coefficient (DSC) of 75.37 and a mean Normalized Surface Dice (NSD) of 81.67.

Our contributions are threefold: (1) We propose a nnU-Net-based framework for UDA to enhance the performance of existing methods. (2) We introduce a three-stage pipeline to improve both effectiveness and efficiency. (3) We present

a method for selecting robust pseudo labels using both a universal segmentation model and an MR-specific model.

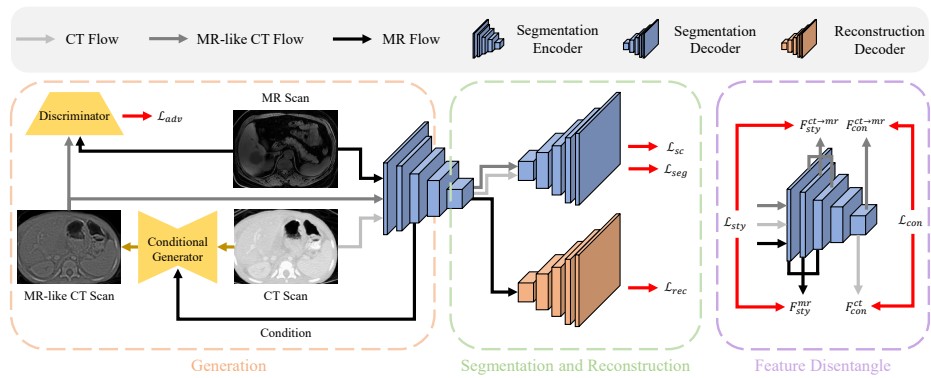

**Fig. 1.** Overview of our proposed nnU-Net-based UDA framework including generation, segmentation and reconstruction, and feature disentangle.

## 2 Method

Let $\mathbb{D}^{ct} = \{\mathcal{X}_i^{ct}, \mathcal{Y}_i^{ct}\}_{i=1}^{N^{ct}}$ and $\mathbb{D}^{mr} = \{\mathcal{X}_i^{mr}\}_{i=1}^{N^{mr}}$ denote the sets of labeled CT scans and unlabeled MR scans, respectively, where $\mathcal{X}_i^* \in \mathbb{R}^{H \times W \times D}$ represents the $i$-th scan and $Y_i^*$ its corresponding label. In this study, we propose a three-stage pipeline for accurate MR scan segmentation, comprising the UDA phase, pseudo-labeling phase, and efficiency phase. We detail each phase below.

**Table 1.** Detailed Configurations of the models.

| Settings | Uni-Net | UDA Generator | MR-Net | LW-Net |
|---|---|---|---|---|
| Base channels | 32 | 32 | 32 | 16 |
| convolution numbers | 3 | 2 | 3 | 2 |
| Downsampling times | 5 | 4 | 5 | 4 |
| Patch size | (48, 224, 224) | (48, 224, 224) | (48, 224, 224) | (32, 128, 192) |
| Input spacing | (2.5, 0.8, 0.8) | (2.5, 0.8, 0.8) | (2.5, 0.8, 0.8) | (4.0, 1.2, 1.2) |

### 2.1 UDA Phase

In this phase, we aim to train a robust universal segmentation model $\mathbb{S}$, called Uni-Net, using the proposed UDA framework, as illustrated in Fig. 1. To enhance

representation learning on MR scans, we improve the vanilla nnU-Net with a reconstruction decoder and a reconstruction loss $\mathcal{L}_{rec}$ based on mean squared error, which has the same architecture as the segmentation decoder.

For domain alignment, we employ a triple-level alignment strategy from [2]. For input alignment, as Fourier-based image translation [18] is unsuitable for CT-to-MR translation, we design a conditional generator $\mathbb{G}$ based on CGAN [15]. The condition is the MR features extracted by the 3rd blocks of the segmentation encoder. The generated MR-like CT scan $\mathcal{X}^{ct \to mr}$ and the MR scan $\mathcal{X}^{mr}$ are fed into a patch-based discriminator $\mathbb{D}$ [16] to calculate the adversarial loss $\mathcal{L}_{adv}$ based on cross-entropy loss. For feature alignment, we first disentangle features following [5]. Low-level features extracted by 1st, 2nd, and 3rd blocks of the segmentation encoder are regarded as style features $F_{sty}$, while high-level features from the last block are considered content features $F_{con}$. We then apply a style consistency constraint $\mathcal{L}_{sty}$ on the style features of $\mathcal{X}^{ct \to mr}$ and $\mathcal{X}^{mr}$, and a content consistency constraint $\mathcal{L}_{con}$ on the content features of $\mathcal{X}^{ct}$ and $\mathcal{X}^{ct \to mr}$, following [2]. For output alignment, besides the segmentation loss $\mathcal{L}_{seg}$, we apply a segmentation consistency constraint $\mathcal{L}_{sc}$ to the segmentation predictions of CT scans and MR-like CT scans. The detailed configurations of the generator and Uni-Net are provided in Table 1. The discriminator $\mathbb{D}$ is trained with the traditional adversarial loss:

$$\min_{\mathbb{D}} \mathcal{L}_{adv}(\mathbb{D}(\mathcal{X}^{ct \to mr}), 0) + \mathcal{L}_{adv}(\mathbb{D}(\mathcal{X}^{mr}), 1). \tag{1}$$

The generator $\mathbb{G}$ and Uni-Net $\mathbb{S}$ are trained using a warm-up mechanism for improved convergence. During the first 25 epochs, we optimize:

$$\min_{\mathbb{G}} \mathcal{L}_{adv}(\mathbb{D}(\mathcal{X}^{ct \to mr}), 1), \quad \min_{\mathbb{S}} \mathcal{L}_{seg}(\mathbb{S}(\mathcal{X}^{ct}), \mathcal{Y}^{ct}) + 0.1 * \mathcal{L}_{rec}. \tag{2}$$

After warm-up, the overall objective is:

$$\min_{\mathbb{G}, \mathbb{S}} \mathcal{L}_{adv}(\mathcal{X}^{ct \to mr}, 1) + \beta \mathcal{L}_{seg}(\mathbb{S}(\mathcal{X}^{ct}), \mathcal{Y}^{ct}) + \alpha \mathcal{L}_{seg}(\mathbb{S}(\mathcal{X}^{ct \to mr}), \mathcal{Y}^{ct}) + \tag{3}$$
$$0.1 * \alpha \mathcal{L}_{con} + 0.01 * \alpha \mathcal{L}_{sty} + 0.1 * \mathcal{L}_{rec},$$

where $\alpha$ and $\beta$ are weights respectively controlling the training extent on MR-like CT scans and CT scans, which are computed as:

$$\alpha = min(1, (epoch/250)^{0.5}), \tag{4}$$

$$\beta = \begin{cases} max(0.5, 1 - ((epoch - 500)/250)^2), \ epoch > 500 \\ 1, \ epoch \leq 500 \end{cases}. \tag{5}$$

We progressively increase the model's focus on MR-like CT scans while reducing its reliance on CT scans.

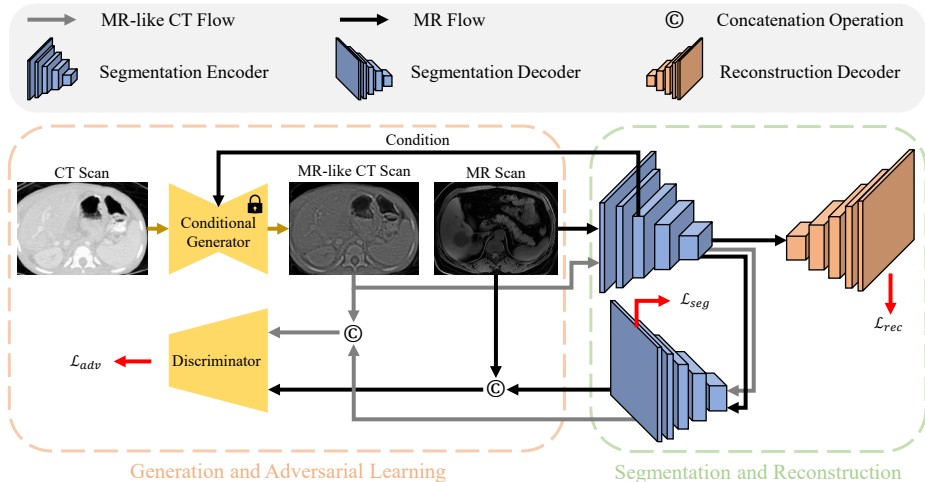

**Fig. 2.** Overview of the fine-tuning process, including generation and adversarial learning, and segmentation and reconstruction. During inference, MR scans will be fed into the segmentation encoder and decoder to produce predictions.

## 2.2   Pseudo-Labeling Phase

After establishing a strong baseline $\mathbb{S}$, we use it to generate pseudo labels $\mathcal{P}^{mr}$ for unlabeled MR scans. For the AMOS dataset, $\mathbb{S}$ is applied directly to the unlabeled MR data. For the LLD-MMRI dataset, we first register scans from eight modalities per patient following [1] and perform inference on four modalities (*i.e.*, C+A, C+Delay, C+V, and C-pre) that exhibit styles similar to CT. We then combine the predictions of these modalities by voting to generate pseudo labels.

Subsequently, we train a large nnU-Net, referred to as MR-Net, on the MR scans with pseudo labels using the default nnU-Net training procedure. The configurations of MR-Net are detailed in Table 1. This trained model generates more accurate pseudo labels $\tilde{\mathcal{P}}^{mr}$. We then freeze the generator $\mathbb{G}$ and fine-tune the Uni-Net $\mathbb{S}$ as follows:

$$\min_{\mathbb{S}} \mathcal{L}_{seg}(\mathbb{S}(\mathcal{X}^{ct\to mr}), \mathcal{Y}^{ct}) + \mathcal{L}_{seg}(\mathbb{S}(\mathcal{X}^{mr}), \tilde{\mathcal{P}}^{mr}) + \tag{6}$$
$$0.1 * \mathcal{L}_{adv}(\mathbb{D}'(Cat(\mathcal{X}^{mr}, \mathbb{S}(\mathcal{X}^{mr}))), 1) + 0.1 * \mathcal{L}_{rec},$$

$$\min_{\mathbb{D}'} \mathcal{L}_{adv}(\mathbb{D}'(Cat(\mathcal{X}^{ct\to mr}, \mathbb{S}(\mathcal{X}^{ct\to mr}))), 1) + \tag{7}$$
$$\mathcal{L}_{adv}(\mathbb{D}'(Cat(\mathcal{X}^{mr}, \mathbb{S}(\mathcal{X}^{mr}))), 0),$$

where $Cat$ denotes concatenation, and $\mathbb{D}'$ is a discriminator similar to $\mathbb{D}$ but with a different number of input channels. The fine-tuning process is illustrated in Fig. 2. We use the fine-tuned Uni-Net $\mathbb{S}$ to generate pseudo labels $\acute{\mathcal{P}}^{mr}$. The uncertainty

$$u = \frac{\sum \tilde{\mathcal{P}}^{mr} \neq \acute{\mathcal{P}}^{mr}}{\sum \tilde{\mathcal{P}}^{mr} > 0} \tag{8}$$

is calculated to select robust pseudo labels $\hat{\mathcal{P}}^{mr}$ from $\tilde{\mathcal{P}}^{mr}$, with empirical thresholds set at 0.25 and 0.5 for the LLD-MMRI and AMOS datasets, respectively.

### 2.3   Efficiency Phase

Inspired by [8], we further train a lightweight nnU-Net, called LW-Net, to enhance segmentation efficiency during deployment. This model is trained exclusively on MR scans with selected pseudo labels $\hat{\mathcal{P}}^{mr}$, following the default training process in nnU-Net. Detailed configurations of LW-Net are provided in Table 1. We employ smaller patch sizes and input spacings to reduce computational complexity and decrease model parameters by using fewer base channels, convolutions, and downsampling layers. The trained LW-Net is then used for final testing to balance performance and efficiency.

**Table 2.** Development environments and requirements.

| | |
|---|---|
| System | Ubuntu 18.04.5 LTS |
| CPU | Intel(R) Xeon(R) CPU E5-2690 v3 @ 2.60GHz |
| RAM | 16×64GB; 3200MT/s |
| GPU (number and type) | Two NVIDIA GeForce RTX 3090 24G |
| CUDA version | 11.7 |
| Programming language | Python 3.10 |
| Deep learning framework | PyTorch 2.0.1, Torchvision 0.15.2 |
| Specific dependencies | nnU-Net 1.7.0 |
| Code | https://github.com/Chen-Ziyang/FLARE2024-Task3 |

## 3   Experiments

### 3.1   Dataset and evaluation measures

The FLARE2024 dataset expands upon the FLARE2022 dataset [13] by incorporating additional unlabeled MR scans from the LLD-MMRI[1] and AMOS [12] datasets. The training set includes 2,050 labeled CT scans with annotations for

---

[1] https://zenodo.org/records/7852363

**Table 3.** Training protocols of UDA and fine-tuning.

| | |
|---|---|
| Network initialization | "He" normal initialization |
| Batch size | 2 |
| Patch size | 48×224×224 |
| Total epochs | 1000 (UDA) / 100 (fine-tuning) |
| Optimizer | SGD with nesterov momentum ($\mu = 0.99$) |
| Initial learning rate (lr) | 0.01 |
| Lr decay schedule | $lr = 0.01 \times (1 - epoch/1000)^{0.9}$ |
| Training time | 72.3 hours (UDA) / 7.2 hours (fine-tuning) |
| Segmentation loss function | Dice loss and cross entropy loss |
| Number of model parameters | $\mathbb{S}$ (119.3M) $\mathbb{G}$ (16.5M) $\mathbb{D}$ (2.6M) |
| Number of flops | $\mathbb{S}$ (1149G) $\mathbb{G}$ (716G) $\mathbb{D}$ (24G) |
| CO$_2$eq | 171.7 Kg (UDA) / 17.2 Kg (fine-tuning) |

**Table 4.** Training protocols of MR-Net and LW-Net.

| | |
|---|---|
| Network initialization | "He" normal initialization |
| Batch size | 4 (MR-Net) / 16 (LW-Net) |
| Patch size | 48×224×224 (MR-Net) / 32×128×192 (LW-Net) |
| Total epochs | 1000 (MR-Net) / 200 (LW-Net) |
| Optimizer | SGD with nesterov momentum ($\mu = 0.99$) |
| Initial learning rate (lr) | 0.01 |
| Lr decay schedule | $lr = 0.01 \times (1 - epoch/1000)^{0.9}$ |
| Training time | 63.1 hours (MR-Net) / 39.5 hours (LW-Net) |
| Loss function | Dice loss and cross entropy loss |
| Number of model parameters | 81.0M (MR-Net) / 5.4M (LW-Net) |
| Number of flops | 1552G (MR-Net) / 1088G (LW-Net) |
| CO$_2$eq | 123.1 Kg (MR-Net) / 54.0 Kg (LW-Net) |

13 organs and 4,818 unlabeled MR scans. The validation set includes 100 MR scans, and the testing set includes 300 MR scans. We evaluate our models using three measures: the Dice Similarity Coefficient (DSC) and Normalized Surface Dice (NSD) for accuracy, and running time for efficiency.

### 3.2   Implementation details

Development environments and requirements are summarized in Table 2. Training protocols for UDA and fine-tuning are outlined in Table 3, while those for MR-Net and LW-Net are detailed in Table 4.

**Training protocols.** We apply on-the-fly data augmentation techniques, including additive brightness, gamma correction, rotation, scaling, and elastic deformation during training. No test-time augmentation (TTA) is used during inference. The final model is empirically selected as the optimal one. For LW-Net, although the model is trained for 1000 epochs, the 200th epoch model is chosen for testing.

**Table 5.** Quantitative evaluation results of the Uni-Net, fine-tuned Uni-Net, MR-Net, and LW-Net. RK: Right Kidney. IVC: Inferior Vena Cava. RAG: Right Adrenal Gland. LAG: Left Adrenal Gland. LK: Left Kidney.

| Target | Uni-Net | | fine-tuned Uni-Net | | MR-Net | | LW-Net | |
|---|---|---|---|---|---|---|---|---|
| | DSC(%) | NSD(%) | DSC(%) | NSD(%) | DSC(%) | NSD (%) | DSC(%) | NSD (%) |
| Liver | 85.96 | 83.65 | 88.41 | 87.47 | 90.85 | 90.73 | 90.83 | 91.07 |
| RK | 85.82 | 83.99 | 88.20 | 85.78 | 87.21 | 84.95 | 87.68 | 86.38 |
| Spleen | 74.85 | 75.86 | 82.30 | 83.38 | 87.58 | 88.60 | 85.34 | 87.15 |
| Pancreas | 70.24 | 80.86 | 75.07 | 86.15 | 79.66 | 90.31 | 78.41 | 90.67 |
| Aorta | 83.49 | 86.35 | 85.33 | 87.89 | 88.23 | 90.13 | 88.21 | 92.02 |
| IVC | 60.45 | 62.25 | 67.18 | 68.31 | 69.67 | 70.83 | 71.25 | 72.83 |
| RAG | 59.82 | 76.13 | 62.34 | 78.95 | 61.17 | 77.04 | 59.36 | 76.47 |
| LAG | 66.24 | 79.28 | 68.15 | 81.24 | 65.63 | 78.24 | 62.57 | 77.03 |
| Gallbladder | 70.86 | 69.36 | 75.07 | 72.08 | 69.24 | 64.73 | 69.47 | 63.23 |
| Esophagus | 51.75 | 63.73 | 57.42 | 69.59 | 60.79 | 73.95 | 61.87 | 77.49 |
| Stomach | 72.11 | 73.96 | 76.85 | 79.40 | 80.29 | 82.83 | 78.39 | 81.21 |
| Duodenum | 57.99 | 75.74 | 61.05 | 78.29 | 60.06 | 80.11 | 58.06 | 79.59 |
| LK | 84.29 | 83.00 | 87.79 | 85.15 | 89.47 | 86.42 | 88.35 | 86.53 |
| Average | 71.07 | 76.47 | 75.01 | 80.28 | 76.14 | 81.45 | 75.37 | 81.67 |

## 4    Results and Discussion

### 4.1    Quantitative results on validation set

We evaluate the Uni-Net, fine-tuned Uni-Net, MR-Net, and LW-Net on the validation set, covering 13 organs. Results are presented in Table 5. Key observations include: (1) All models perform well on easier categories such as the liver and right kidney, but performance declines on more challenging categories like the esophagus and duodenum; (2) Utilizing pseudo labels for fine-tuning or training a model from scratch improves performance; (3) Our LW-Net achieves a competitive overall DSC and the best overall NSD metric.

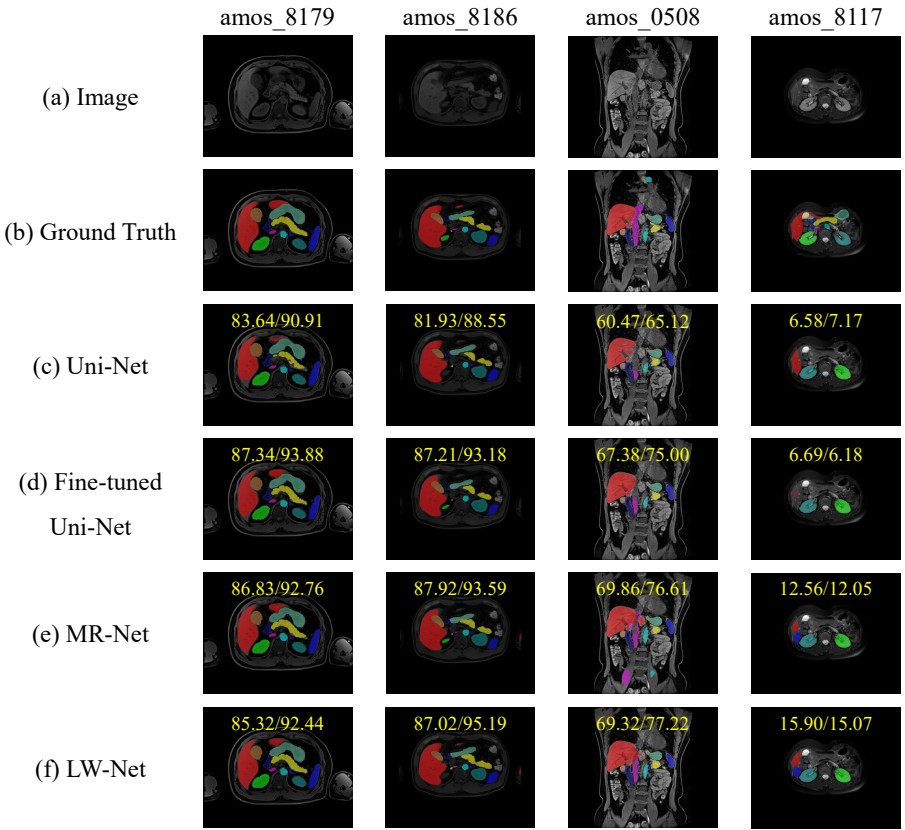

**Fig. 3.** Qualitative results of the Uni-Net, fine-tuned Uni-Net, MR-Net, and LW-Net on two easy cases (amos_8179 and amos_8186) and two hard cases (amos_0508 and amos_8117). The DSC/NSD metrics are shown in the top of each image.

### 4.2    Qualitative results on validation set

Fig. 3 presents four representative segmentation results from the Uni-Net, fine-tuned Uni-Net, MR-Net, and LW-Net. The cases include two easy examples (amos_8179 and amos_8186) and two challenging ones (amos_0508 and amos_8117). For amos_8179 and amos_8186, all models successfully segment most organs. In contrast, for amos_0508 and amos_8117, noticeable under-segmentation and over-segmentation errors are evident. These issues are likely due to domain shifts caused by the distinct styles of these cases compared to the CT modality. Despite these challenges, LW-Net delivers competitive results while maintaining high segmentation efficiency.

**Table 6.** Segmentation efficiency results of the Uni-Net, fine-tuned Uni-Net, MR-Net, and LW-Net.

| Target | Uni-Net | fine-tune | MR-Net | LW-Net |
|---|---|---|---|---|
| Total Running Time (s) | 3333.17 | 818.79 | 1187.17 | 479.30 |

**Table 7.** Results of the LW-Net on the testing set.

| Method | DSC(%) | NSD(%) | Time (s) | GPU |
|---|---|---|---|---|
| LW-Net | $37.5 \pm 26.9$ | $37.6 \pm 29.6$ | $15.4 \pm 4.0$ | $856590.4 \pm 238940.1$ |

### 4.3    Segmentation efficiency results on validation set

Table 6 reports the segmentation efficiency results on the validation set, using the official nnU-Net evaluation code and a NVIDIA GeForce RTX 3090 GPU. The results demonstrate that LW-Net exhibits superior segmentation efficiency compared to other models.

### 4.4    Limitation and future work

The proposed UDA method involves a complex three-part framework, which complicates the evaluation of each component's contribution. Additionally, extending the framework to other tasks requires extensive manual tuning, which is time-consuming and labor-intensive. Future work will focus on simplifying the framework to enhance its extensibility.

## 5    Conclusion

In this paper, we present an enhanced approach to utilizing nnU-Net for cross-modality UDA and introduce a three-stage pipeline specifically designed for abdominal organ segmentation in MR scans. In the first stage, we establish an universal segmentation model using the nnU-Net-based UDA framework, incorporating a triple-level alignment strategy. In the second stage, we generate and select robust pseudo labels for unlabeled MR scans to improve model training. In the third stage, we train a lightweight nnU-Net using the selected MR scans with pseudo labels to balance the efficiency and effectiveness. We believe that our proposed pipeline provides a strong baseline for abdominal organ segmentation in MR scans and can be extended to other UDA tasks.

**Acknowledgements**  This work was supported in part by Shenzhen Science and Technology Program under Grants JCYJ20220530161616036, in part by the National Natural Science Foundation of China under Grant 62171377 and Grant 92470101, and in part by the Innovation Foundation for Doctor Dissertation of Northwestern Polytechnical University under Grant CX2022056. We thank all data owners for making the medical images publicly available and CodaLab [17] for hosting the challenge platform.

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
