# OpenReview forum: "Rethinking nnU-Net for Cross-Modality Unsupervised Domain Adaptation in Abdominal Organ Segmentation"
_MICCAI.org/2024/Challenge/FLARE — FLARE 2024 withMinorRevisions_

### Official Review · Reviewer_5M9C · 2025-01-17
**Review Comments**

**Rating:** 9
**Confidence:** 5

**Review:**

The paper introduces a universal segmentation model (Uni-Net) by aligning tasks at the input, feature, and output levels. It fine-tunes Uni-Net and selects labels from both Uni-Net and MR-Net outputs to train the lightweight LW-Net, balancing segmentation accuracy and performance. However, there are a few areas for improvement:

 **1.Baseline Experiment**: A baseline experiment comparing the proposed Uni-Net with a network trained directly using CT labels would strengthen the evidence for the effectiveness of Uni-Net.

**2. Clarity in Figure 2**: In Figure 2, it would be helpful to clearly differentiate the network structures used for training and inference. Alternatively, the figure caption could specify the modules (such as the encoder and segmentation decoder) used during inference, providing greater clarity.

Despite these minor suggestions for improvement, the paper provides valuable contributions to the field and is well-executed overall.

---

> ### Author Response · Authors · 2025-03-29
>
> Thanks for your valuable comments.
> Since the competition organizer will provide the baseline results for comparison, we have omitted it.
> We have addressed comment 2 in our revised manuscript.

---

### Official Review · Reviewer_tEF3 · 2025-01-28
**comments**

**Rating:** 8
**Confidence:** 5

**Review:**

This paper proposes an nnU-Net-based three-stage pipeline for cross-modality unsupervised domain adaptation (UDA) in abdominal organ segmentation, aiming to transfer knowledge from CT to MR scans. The approach introduces a triple-level alignment strategy, pseudo-labeling, and a lightweight nnU-Net to improve segmentation accuracy and efficiency, achieving promising results on FLARE2024 Task 3.
However, some aspects require further refinement:
1. The three-stage pipeline, while effective, is complex and requires extensive computational resources, limiting its practicality in real-world applications.
2. The process for selecting pseudo-labels lacks detailed analysis on uncertainty estimation and error propagation.

---

> ### Author Response · Authors · 2025-03-29
>
> Thank you for your valuable comments. Here are our responses.
> 1) Although the three-stage pipeline is complex, the used GPU resources and training time are still acceptable.
> 2) Although there may have been errors introduced in the pseudo label, it still brings performance gains.

---

### Official Review · Reviewer_AbV6 · 2025-03-02
**Minor writing issues**

**Rating:** 7
**Confidence:** 5

**Review:**

“ Authors Suppressed Due to Excessive Length” Please shorten author abbr. by using et al.
`FLARE2024 Task3` is not a good method abbreviation. Please change it to a short name of your method

Table 1: format issue: add a horizontal line to the head and assign the numbers in the middle
All figures with CT images: adjust window level and width to 40/400
Fig. 1-2. Add more explanations to the network

---

> ### Author Response · Authors · 2025-03-29
>
> We have addressed the reviewers' comments in revised manuscript.

---

### Decision · Program_Chairs · 2025-03-20

**Decision:**

Accept

**Comment:**

Minors:

 Sec. 4.3 subtitle should be ``Segmentation efficiency results on testing set" not validation set. Also fix the Table index.

---

> ### Author Response · Authors · 2025-04-02
>
> We have addressed all the comments in revised manuscript.